# From Data to Model: Anomaly Detection of 3D GPR Data in CuDeRes Model Space

## Abstract

Ground Penetrating Radar (GPR) offers in-depth subterranean insights, yet subsurface anomaly detection in GPR data remains challenging due to limited training data, typically confined to some normal data samples free from any subsurface structures or anomalies, and the variability of subsurface conditions. In response, this paper introduces practical and accurate subsurface anomaly detection within the Cubic Decay Reservoir Network (CuDeRes) model space. Our approach employs commonly available normal GPR data, segmented into blocks. Each data block is independently fitted using the introduced CuDeRes, which incorporates three reservoirs with spatial decay to adequately capture the data-inherent multidirectional dynamics, resulting in a compact fitted readout model. Representing each data block with the fitted model, together with the distance measurement between models, the original GPR data blocks are mapped into the CuDeRes model space, and the fitted models are collected into a "Model Depot". For subsequent anomaly detection in newly collected GPR data, the same segmentation and CuDeRes fitting approaches are applied, where the data blocks are represented by fitted models for comparative assessment against the model depot. Anomalies are detected through model dissimilarities, and subsequently clustered within the CuDeRes model space, allowing us to accurately identify the data blocks with potential subsurface anomalies and ascertain their anomaly types. Experiments on real-world GPR data demonstrate the practical effectiveness of our approach, notably using only limited normal data.

## 1 Introduction

Ground Penetrating Radar (GPR), employing Electromagnetic (EM) waves, serves as a vital tool for detecting subsurface structures or anomalies beneath urban roadways (Chen & Cohn, 2011). This technology operates by transmitting high-frequency waves and analyzing the signals reflected back, which reveal the diverse properties and compositions under the surface (Zhou et al., 2018). The integration of multi-channel technology in GPR systems facilitates the simultaneous collection of EM waves across multiple antennas, enabling the generation of 3-dimensional GPR (3D GPR) data[1], essential for assessing subsurface conditions (Goodman et al., 2013).

Detecting underground anomalies in GPR data typically involves segmenting data along the detecting direction, and further identifying sections with potential subterranean issues (Zhou et al., 2023), a labor-intensive and time-consuming process when performed manually. Algorithms for image or signal feature extraction and classification aid in categorizing GPR data, but struggle due to the variable characteristics of subsurface anomalies, affected by their composition, size, and surrounding environment. Recent advances in Deep Learning (DL), particularly Convolutional Neural Networks (CNNs), have also been applied to object and anomaly detection in GPR data (Liu et al., 2021; Liang et al., 2022b). Despite their potential, DL methods face significant considerations: 1) The scarcity of GPR data, particularly in a targeted detection area, often leads to limited training datasets predominantly composed of normal samples; 2) The variability of underground environments undermines the generalization capabilities of DL models, restricting their adaptability to various or unfamiliar

---

[1]For simplicity, unless specifically stated otherwise, the term "GPR data" used in this paper refers to 3D GPR data. More description about this data is provided in Section 2 Related Work.

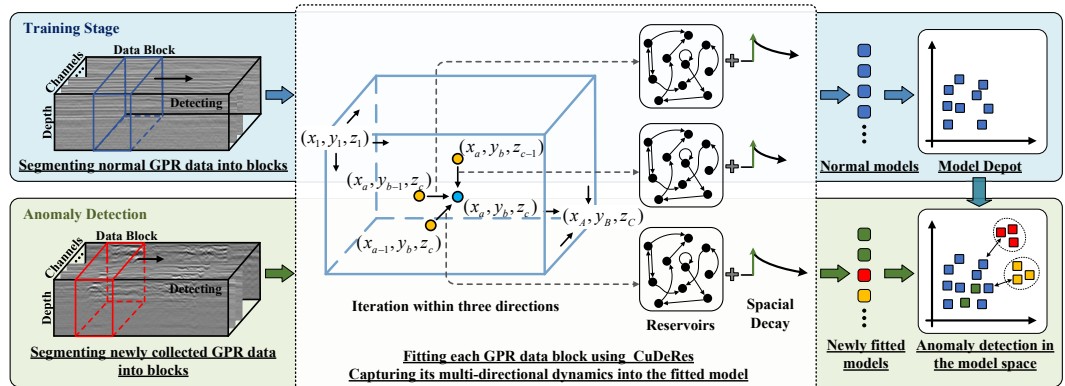

Figure 1: In the training stage, normal GPR data obtained in the detecting area is segmented into same-size blocks, each fitted by CuDeRes to capture its multi-directional dynamics, with the fitted models collected into the "Model Depot". For subsequent anomaly detection, the same segmentation and fitting approaches are applied to newly collected GPR data, with fitted models compared against the model depot. Anomalies could be identified based on model dissimilarities, and then clustered.

subsurface conditions; 3) The inherent complexity and extensive parameterization of DL models, especially those based on 3D-CNNs, necessitate substantial computational resources.

Given the challenges associated with GPR data processing, the framework of Learning in the Model Space (LMS) offers a viable alternative (Chen et al., 2013). LMS transitions data from data space to model space by fitting the data with appropriate models that capture and describe the dynamics (i.e., changing information) within the data. Consequently, the fitted models serve as more stable and parsimonious representations of the data, enabling effective implementation of learning algorithms on the models rather than the original data. Successfully applied to diagnosing the Barcelona water network (Quevedo et al., 2014) and the Tennessee Eastman Process (Chen et al., 2014), along with diverse time-series classification tasks (Gong et al., 2018; Wu et al., 2022), LMS has proven its efficacy in various tasks using the Echo-State-Network-based (ESN-based) methods for sequential data fitting and representation. Notably, LMS focuses on the data-intrinsic dynamics, which allows for reduced reliance on training data and diminishes the computational demands compared to many DL methodologies, particularly when optimally configuring the model (Ma et al., 2020).

While efforts have been made to apply LMS in GPR data processing, the following considerations persist: 1) LMS typically fits data uni-directionally to capture inherent dynamics, primarily designed for sequential data with contextual relationships. However, GPR data exhibits both vertical variations and horizontal correlations, stemming from the continuity of the subsurface medium and EM waves. Effectively capturing these multi-directional dynamics is essential for accurate fitting, representation, and classification of GPR data. 2) According to the spacing between GPR antenna channels, the sampling frequency along the detecting direction, and the dielectric constant of the underground medium, the distances between adjacent points vary across different directions within GPR data, leading to differing correlations along distinct spatial orientations, with some directions correlating stronger than others. Such variation necessitates a precise capture of the unique dynamics within each direction. Despite attempts to optimize dynamic capture using accurately labeled multi-type data, acquiring a sufficiently diverse training dataset often remains impractical.

Addressing the above, this paper introduces learning in the Cubic Decay Reservoir Network (CuDeRes) model space for subsurface anomaly detection in GPR data, illustrated in Figure 1. Our approach only requires some normal GPR data, commonly available from the detecting area, to support anomaly detection in subsequently collected data. We segment the normal GPR data into blocks, with data segments in each block, referred to as GPR data blocks, being fitted by CuDeRes[2]. Given GPR data's vertical continuity along EM waves and horizontal correlations due to subsurface medium consistency, each point in GPR data is correlated with its surroundings in multiple direc-

---

[2]In this paper, "CuDeRes" designates the network used for fitting GPR data, resulting in the "CuDeRes fitted readout model" for data representation, also simplified as the "CuDeRes model" or "fitted model".

tions. Different underground structures manifest different dynamics in the GPR data. The proposed CuDeRes integrates three reservoirs and applies spatial decay in each direction, constructing connections between points within the data across multiple directions, during which it strengthens the correlation with nearer points while weakening it with those further away. Fitting the data block with CuDeRes effectively captures the multi-directional dynamics within the GPR data, resulting in a compact fitted readout model. Representing each block with the fitted model, coupled with the distance measurement between models, transitions the original GPR data blocks into the CuDeRes model space. These models, derived from normal data blocks, are collected into a "Model Depot".

For anomaly detection in subsequent GPR data, we continue with the same segmentation process and fit each block with CuDeRes, deriving the fitted model for each data block. Given the consistent dynamics within GPR data, blocks originating from identical subsurface structures derive similar CuDeRes models, whereas models fitted from diverse subsurface structures manifest significant variations, depicting the unique dynamics captured. Each newly fitted model is then evaluated against the established model depot, obtaining its anomaly score. Models registering higher anomaly scores, indicative of potential anomalies, are identified and then grouped, allowing us to precisely identify the corresponding abnormal block and determine the type of anomaly associated with each identified abnormal GPR data block. The main contributions of this paper are as follows:

- The introduced CuDeRes incorporates three reservoirs with spatial decay at each direction, enhancing correlations with nearer data points and diminishing those with distant ones, adequately and accurately capturing multi-directional dynamics within GPR data.

- Representing GPR data with the compact fitted CuDeRes model, coupled with the directly computable distance measurement between models, allows for further anomaly detection to be effectively performed within the category-discriminative CuDeRes model space.

- Our approach focuses on the inherent dynamics present in GPR data, and leverages only limited normal GPR data, easily obtainable in the detecting area, to support the subsequent anomaly detection, enabling its practical usability in real-world settings.

## 2 RELATED WORK

### 2.1 GPR DATA ANALYZING

GPR data, specifically the multi-channel 3D GPR data, provides an advanced tool for viewing what lies beneath the surface. As depicted in Figure 2, unlike single-channel GPR which provides two-dimensional data, multi-channel GPR systems feature arrays of antennas that simultaneously send and receive EM waves across multiple channels.

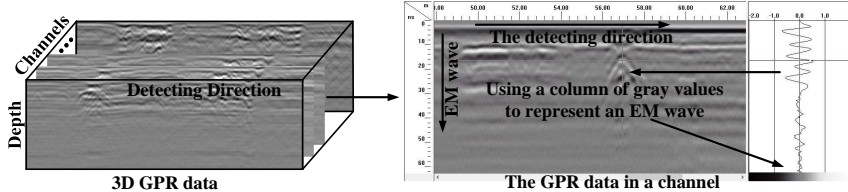

Figure 2: Single-channel GPR data could be visualized in image format, organizing received EM waves horizontally by time or space, with wave intensities shown as grayscale values. Due to EM wave refraction and reflection, GPR data may not directly reflect actual subsurface structures, requiring further analysis. In 3D GPR data, critical changing information is presented both along and among EM waves, horizontally and vertically with different scales in distinct directions.

Although 2D GPR data could be displayed in image format, and 3D GPR data as image sequences, essentially, GPR data comprises the collection, arrangement, and representation of EM waves. Each data point reflects the intensity of EM waves at a specific subsurface location. Given the continuity of underground media and the presence of anomalies, GPR data display valuable changing information not only in the vertical direction along the EM waves but also along the detection path and across various channels. Moreover, the scales differ across various directions within GPR data, resulting in

variable correlations between each point and its adjacent ones in different directions. For example, the distance between two adjacent channels could be approximately 15cm, whereas the gap between two rows of EM waves within a channel ranges from 2 to 6cm. Additionally, the spacing between two points within a row, affected by the dielectric constant of the subsurface medium, spans from 2 to 5cm. Accurately capturing the multi-directional dynamics inherent in GPR data also necessitates considering these direction-wise scale differences.

Recent advancements built on DL have mainly delineated two strategies for 3D GPR data analysis. 1) The first entails direct processing of 3D data: Liu et al. (2022) introduced a data augmentation technique called Multiple Mirror Encoding (MME) to accommodate 3D GPR data, employing the C3D network (Tran et al., 2015) to facilitate spatio-temporal feature extraction; Similarly, the 3DInvNet leverages a 3D CNN equipped with a feature attention mechanism to mitigate noise, followed by a U-shaped encoder-decoder architecture that incorporates multiscale feature aggregation to generate detailed underground permittivity maps. 2) The second extracts single-channel or cross-sectional profiles from the 3D data, which are subsequently analyzed and integrated: For instance, UcNet (Kang et al., 2019) merges CNN with phase analysis to enhance the resolution of GPR data; Another research (Liang et al., 2022a) investigated and compared the VGG and ResNet frameworks, focusing on their application in classifying GPR datasets containing subterranean anomalies. For 3D GPR data analysis, DL approaches face critical considerations: the scarcity of multi-type labeled GPR data, often results in limited training data predominantly composed of normal samples in targeted detection areas; the complex and parameter-heavy network, especially the 3D CNNs, while analyzing channels individually also risks neglecting the inter-channel dynamics.

## 2.2 LMS and Its Application in GPR Data Processing

The LMS framework was initially introduced in (Chen et al., 2013) and utilized for fault diagnosis in sequential data. Typically, LMS methods use Echo State Networks (ESN) to fit individual data instances. Each data instance, represented by its respective fitted model, is transitioned into the model space, with a defined distance measurement between models. Learning algorithms could then be effectively performed within this model space, leveraging the dynamics captured from the data. The adaptability of LMS has been expanded to include applications in time series classification, disease diagnosis (Bianchi et al., 2020), and addressing concept drift (Chiu & Minku, 2022).

Applying LMS to GPR data processing, prior studies (Zhou et al., 2023; Chen et al., 2024) have explored anomaly detection of 2D GPR data using an enhanced ESN-based network, capturing horizontal and vertical dynamics respectively into fitted models for further classification. For 3D GPR data, characterized by higher dimensions and more complex intrinsic dynamics, a prevalent approach has been to apply LMS independently to each channel, treating each as 2D data, and then aggregating the results (Liu et al., 2024). However, this technique does not account for the dynamics among channels. Build-

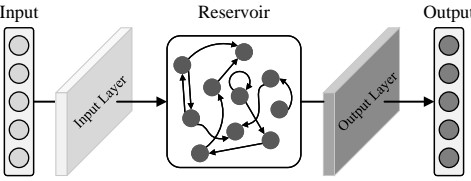

Figure 3: Typically, an ESN primarily consists of the input layer, a hidden layer containing a reservoir, and the output layer.

ing upon the above, Zhou et al. (2024) proposed augmenting the ESN's hidden layer with three reservoirs to better capture multi-directional dynamics in 3D data. Despite this enhancement, this modification does not fully address the spatial scale variations across various directions within GPR data. While this study also recommended using labeled data to optimize the fitting accuracy and classification of fitted models, as introduced above, the scarcity of sufficient, diverse, and accurately labeled GPR data in specific detecting areas greatly restricts its practical application.

## 2.3 A Brief Introduction of Echo State Network

Echo State Networks (ESNs) represent a subclass of Recurrent Neural Networks (RNNs), known for simplicity and efficiency in processing sequential data. As depicted in Figure 3, an ESN's architecture consists of three primary components: an input layer, a reservoir within the hidden layer, and an output layer. A notable feature of ESNs is the use of randomized and fixed weights for both the input and the reservoir weights, also essential to maintain the Echo State Property (ESP).

In fitting sequential data, ESN first computes the hidden state for each point in the input sequence, and the output layer maps this onto the target sequence, using ridge regression to determine output weights. Although effective in fitting data along the processed direction, ESNs fail to adequately capture dynamics in multi-dimensional data, missing changes in other directions.

## 3 METHODOLOGY

As illustrated in Figure 1, our approach consists of two stages:

- **Training Stage:** Normal GPR data, free from subsurface anomaly and easily available in the detection area, is segmented along the detecting direction into same-size blocks, each independently fitted by CuDeRes to capture its multi-directional dynamics. The fitted models, representing their respective data blocks, are collected into a "Model Depot".

- **Anomaly Detection:** The same segmentation and fitting procedures are employed on newly collected GPR data, with the resulting models compared to those in the model depot. Models that deviate from the norm are identified and clustered, enabling the identification of data blocks that contain subsurface anomalies and the determination of their types.

In our approach, fitting GPR data and capturing its multi-directional dynamics via CuDeRes are crucial for data representation and subsequent anomaly detection on the fitted models. Therefore, we start with a comprehensive introduction to CuDeRes and its fitting process, followed by descriptions of the training stage and anomaly detection in the mode space.

### 3.1 DATA FITTING AND REPRESENTATION VIA CUDERES

#### 3.1.1 FITTING GPR DATA BY CUDERES

Akin to ESN, CuDeRes consists of an input layer, a hidden layer, and an output layer. However, to adequately capture the complex multi-directional dynamics within GPR data, CuDeRes enhances the hidden layer with three reservoirs, each implementing spatial decay, thus effectively building correlations among adjacent points in various directions.

Denoting a GPR data block as $\mathbf{U} \in \mathbb{R}^{A \times B \times C}$, wherein a data point within this block is localized by $(x_a, y_b, z_c)$, and the corresponding value at that point is $u(x_a, y_b, z_c)$. The iteration of CuDeRes begins at the point $(x_1, y_1, z_1)$ and ends at $(x_A, y_B, z_C)$. Sequentially, as shown in Figure 4, each point is sent into the hidden layer, with their hidden states $\mathbf{h} \in \mathbb{R}^{N \times 1}$ calculated as:

$$\mathbf{h}(x_a, y_b, z_c) = g\left(\mathbf{W} \cdot \mathbf{E} \cdot \mathbf{h}^*(x_a, y_b, z_c) + \mathbf{W}^{\text{in}} u(x_a, y_b, z_c)\right), \tag{1}$$

where[3]:

- $a, b, c \in \mathbb{Z}$ represent the indices of the point $(x_a, y_b, z_c)$ in the $x$-, $y$-, and $z$-directions, respectively. $g$ is the activation function $\tanh$. $\mathbf{W}^{\text{in}} \in \mathbb{R}^{N \times 1}$ denotes the input weights.

- $\mathbf{W} \in \mathbb{R}^{N \times 3N}$ contains the reservoir weights for each direction:

$$\mathbf{W} = \begin{bmatrix} \mathbf{W}^x & \mathbf{W}^y & \mathbf{W}^z \end{bmatrix}, \tag{2}$$

  where $\mathbf{W}^x, \mathbf{W}^y, \mathbf{W}^z \in \mathbb{R}^{N \times N}$ represent the reservoir weights along the $x$-, $y$-, and $z$-directions, respectively.

- $\mathbf{E} \in \mathbb{R}^{3N \times 3N}$ represents exponential spatial decay factors applied to each direction:

$$\mathbf{E} = \begin{bmatrix} e^{-\theta(x_a - x_{a-1})}\mathbf{I}_N & 0 & 0 \\ 0 & e^{-\theta(y_b - y_{b-1})}\mathbf{I}_N & 0 \\ 0 & 0 & e^{-\theta(z_c - z_{c-1})}\mathbf{I}_N \end{bmatrix}, \tag{3}$$

  where $\theta > 0$ is the decay rate parameter, $\mathbf{I}_N \in \mathbb{R}^{N \times N}$ represents the identity matrix, $(x_a - x_{a-1})$, $(y_b - y_{b-1})$, and $(z_c - z_{c-1})$ refer to the distances between a point and its

---

[3]The parameters in CuDeRes, denoted by $\mathbf{\Theta} = (\mathbf{W}^{\text{in}}, \mathbf{W}^x, \mathbf{W}^y, \mathbf{W}^z)$, are randomly initialized and fixed. The input weights $\mathbf{W}^{\text{in}}$ span $[-1, 1]$, the reservoirs $\mathbf{W}^x, \mathbf{W}^y, \mathbf{W}^z$ satisfy the Echo State Property (ESP). The initial hidden state $\mathbf{h}(x_0, \cdot, \cdot) = \mathbf{h}(\cdot, y_0, \cdot) = \mathbf{h}(\cdot, \cdot, z_0) = 0$ in each of the three directions.

adjacent points in the $x$-, $y$-, and $z$- direction, respectively. The introduced spatial decay reduces the influence of distant points and assigns greater importance to nearby points, emphasizing local spatial relationships for more accurate dynamic capture.

- $\mathbf{h}^* \in \mathbb{R}^{3N \times 1}$ is the concatenation of the previous hidden states from the three directions:

$$\mathbf{h}^*(x_a, y_b, z_c) = \begin{bmatrix} \mathbf{h}(x_{a-1}, y_b, z_c) \\ \mathbf{h}(x_a, y_{b-1}, z_c) \\ \mathbf{h}(x_a, y_b, z_{c-1}) \end{bmatrix}.$$

During the iteration described in Equation 1 and Figure 4, the hidden state at a point is influenced by both the current point and previous hidden states, creating correlations among neighboring hidden states across three directions. Furthermore, the spatial decay $\mathbf{E}$ appropriately modulates the correlation strength in different directions: correlations between closer data points are amplified, while those between more distant points are diminished. This allows CuDeRes to automatically adapt to varying scales within 3D GPR data, avoiding the interference caused by standardizing data scales. As iterations progress, the ongoing multi-directional correlations form a network, linking each data point to those processed earlier, effectively capturing data-inherent multi-directional dynamics.

After computing the hidden states for all data points within the block, the output value $v$ for each point is calculated from the previous hidden states:

$$v(x_a, y_b, z_c) = \mathbf{W}^{\text{out}} \mathbf{h}^*(x_a, y_b, z_c) + \beta, \quad (4)$$

where $\mathbf{W}^{\text{out}} \in \mathbb{R}^{1 \times 3N}$ denotes the output weights, and $\beta$ is the bias.

The fitting process is accomplished using the "next point prediction" task (Chen et al., 2013). It aims to predict the value of the subsequent point based on processed ones, establishing a mapping between hidden states and corresponding input data points. Explicitly, each output value $v(x_a, y_b, z_c)$, derived from the hidden states, is required to closely match the input $u(x_a, y_b, z_c)$. To achieve this, the output weights $\mathbf{W}^{\text{out}}$ and the bias $\beta$ are determined using ridge regression:

$$\left[\mathbf{W}^{\text{out}} \ \beta\right]^{\mathrm{T}} = (\tilde{\mathbf{H}}\tilde{\mathbf{H}}^{\mathrm{T}} + \lambda^2 \mathbf{I}_{3N})^{-1}\tilde{\mathbf{H}}\mathbf{u}, \quad (5)$$

where $\tilde{\mathbf{H}}$ is the augmentation of the hidden state

Figure 4: The CuDeRes iteration processes data points sequentially, starts at $(x_1, y_1, z_1)$, moves to $(x_A, y_1, z_1)$, then continues from $(x_1, y_2, z_1)$ to $(x_A, y_2, z_1)$, and so forth, until reaching the end $(x_A, y_B, z_C)$. Each point correlates with its predecessors across three directions, with the correlation exponentially diminishing via spatial decay.

matrix $\mathbf{H}$, extended by a row of ones to include bias terms; $\mathbf{H} \in \mathbb{R}^{3N \times ABC}$ is obtained by collecting the previous hidden states of all data points in sequence and column-wise, with each column representing a specific $\mathbf{h}^*(x, y, z)$; $\mathbf{u} \in \mathbb{R}^{ABC \times 1}$ is a vectorized form of the input values $u(x, y, z)$, arranged in the same order as $\mathbf{h}^*(x, y, z)$ in $\mathbf{H}$; $\mathbf{I}_{3N} \in \mathbb{R}^{3N \times 3N}$ represents the identity matrix; and $\lambda$ serves as a regularization.

During the fitting process, CuDeRes's unique iteration adeptly establishes the correlation between adjacent points in the data block, and the spatial decay effectively adjusts the influence across different directions, capturing the data-intrinsic changing information. The fitting approach integrates multi-directional dynamics within the data block into a compact fitted CuDeRes readout model:

$$f(\mathbf{x}) = \mathbf{W}^{\text{out}}\mathbf{x} + \beta. \quad (6)$$

This readout model provides a compact representation of the original data block. When an anomaly arises in a data block, it introduces atypical dynamics, causing the readout model to exhibit distinct compared to those derived from normal GPR data blocks. As a result, representing the data blocks by readout models enhances category discrimination, which in turn improves the effectiveness of model classification compared to using the original data blocks.

### 3.1.2 Distance Measurement between Fitted Models

After fitting the data via CuDeRes, it is crucial to establish a distance to measure the differences between the fitted models, aiding subsequent anomaly detection on the models. Instead of directly

using parameter vectors, which are highly sensitive to specific model parameterizations, the $p$-norm distance (Chen et al., 2013) between models $f_1(\mathbf{x})$ and $f_2(\mathbf{x})$ is adopted:

$$\mathcal{D}_p(f_1, f_2) = \left( \int_C \| f_1(\mathbf{x}) - f_2(\mathbf{x}) \|^p d\mu(\mathbf{x}) \right)^{1/p}. \tag{7}$$

Here, $f_1$ and $f_2$ represent the simplified forms of $f_1(\mathbf{x})$ and $f_2(\mathbf{x})$, respectively; $\mu(\mathbf{x})$ denotes the probability density over the input domain, and the integral range $C$ is $[-1, 1]^{3N}$. For the sake of simplicity, $p$ is set to 2 by default, and we assume $\mathbf{x}$ follows a uniform distribution.

Given the two blocks fitted using CuDeRes, the models derived for each are denoted as follows:

$$\begin{cases} f_1(\mathbf{x}) = \mathbf{W}_1^{\text{out}}\mathbf{x} + \beta_1, \\ f_2(\mathbf{x}) = \mathbf{W}_2^{\text{out}}\mathbf{x} + \beta_2. \end{cases} \tag{8}$$

Substituting Equation 8 into Equation 7 results in:

$$\mathcal{D}_2(f_1, f_2) \propto \frac{1}{3} \| \mathbf{W}_1^{\text{out}} - \mathbf{W}_2^{\text{out}} \|^2 + (\beta_1 - \beta_2)^2. \tag{9}$$

Such direct-measured pair-wise distance measurement specified in Equation 9 facilitates the usage of distance-based learning algorithms on the fitted models[4].

## 3.2 Training Stage: Given Normal Data Only

For the normal 3D GPR data gathered from the detection area, we first segment them into same-size data blocks along the detection direction. This segmentation could be performed using a sliding block technique that progresses along the detection direction (Zhou et al., 2024). Each block is individually fitted by CuDeRes, resulting in a respective readout model. The models obtained from these normal data blocks are collected into a "Model Depot". This process is represented as:

$$\mathcal{M} = \bigcup_{\mathbf{r}_i \in \mathcal{R}} \{ \mathcal{F}(\mathbf{U}) \mid \mathbf{U} \in \mathcal{S}(\mathbf{r}_i) \}, \tag{10}$$

where $\mathcal{R}$ is the collection of normal GPR data samples; $\mathbf{r}_i$ denotes a data sample in $\mathcal{R}$; $\mathcal{S}$ represents the segmentation process; $\mathbf{U}$ is a specific data blocks; and $\mathcal{F}$ refers to CuDeRes fitting process.

## 3.3 Anomaly Detection in CuDeRes Model Space

Given the "Model Depot", anomaly detection on the newly collected GPR data is performed through the following three steps: **1) Data Segmentation and CuDeRes Fitting:** Apply the same segmentation and fitting procedures used during training, obtaining the fitted CuDeRes model for each data block. **2) Model Discrimination:** For each model, find the closest normal model in the model depot through the distance measurement given in Equation 9, examining its anomaly score, with overly high scores indicating anomalies. **3) Model Clustering:** Cluster the abnormal models identified in the previous step, where each cluster signifies a type of anomaly.

The first step, previously described and the same as the training stage, thus would not be repeated here. Details for the subsequent steps are provided as follows.

### 3.3.1 Distance-based Model Discrimination

For anomaly detection with newly collected GPR data, the segmentation and CuDeRes fitting processes generate a set of data blocks, along with their corresponding fitted models. Each model is subjected to a Nearest-Neighbor (NN) search within the model depot to estimate an anomaly score, which helps determine its abnormality.

If an extracted block contains an underground anomaly, owing to the distinct changing information along and among the collected EM waves, there exists differing dynamics in GPR data compared to the normal. As a result, models fitted from such blocks stand out markedly from those fitted from

---

[4]Derivation from Equation 7 to Equation 9 refer to **Appendix**.

normal data, according to the distinct dynamics captured. Specifically, models derived from normal data tend to cluster tightly, while those derived from anomalies are distinctly separated from the normal ones, and positioned far from the normal models.

To assess the anomaly score for a model $f$, the distance to its nearest neighbor in the model depot is computed and the anomaly score is defined as:

$$score(f) = \min_{f^* \in \mathcal{M}} \mathcal{D}_2(f^*, f). \tag{11}$$

A binary classifier then evaluates whether model $f$ is normal or not, based on its anomaly score $score(f)$. If the score exceeds a specified threshold, the model is deemed abnormal; otherwise, it is considered normal.

### 3.3.2 Model Clustering

Through the above steps, models fitted from abnormal data blocks are identified. Different underground anomalies exhibit distinct changing information along and among the collected EM waves, resulting in unique dynamics presented in GPR data. Consequently, models fitted from various types of anomalies show noticeable differences, while blocks segmented from the same type of anomaly derive similar fitted models due to their consistent inherent dynamics. This results in a category-discriminative CuDeRes model space, where clustering[5] is performed to group the identified abnormal models, and each cluster corresponds to a specific type of anomaly.

## 4 Experimental Study

Experiments are conducted on real-world 3D GPR datasets subsequently introduced. All experiments are conducted using Python 3.6 on a desktop with an Intel Core i5-11500 2.70-GHz CPU, 16-GB RAM, and a GeForce RTX 3080Ti 12G graphics card. For the implementation of the proposed method, we initialize the input weights and reservoir weights of CuDeRes randomly following a standard normal distribution. The size of each reservoir is default set as 50, the spectral radii are set to 0.9, the decay rate $\theta$ and the regularization parameter $\lambda$ are set to 1 by default. The threshold of the anomaly score is set to be the average of the pairwise distances between normal models. As for the clusters, we use the official implementation from scikit-learn[6]. To ensure reliability and stability, we report the mean metric under five different random seed settings. The experimental results demonstrate the effectiveness of our approach.

### 4.1 The Utilized 3D GPR data

The GPR data is gathered along cement and asphalt roads, the most prevalent and extensively utilized road types. A 3D GPR system equipped with a 16-channel antenna is utilized. The collected data is segmented into GPR data blocks measuring $16 \times 200 \times 200$, which approximately equates to a physical area of 2.5m (width) $\times$ 4m (detecting direction) $\times$ 4m (depth). Each data block, except for the normal ones, contains an anomaly. The number of blocks for each category is given in Table 1.

Three major types of subsurface anomalies are observed: cavities, looseness, and cracks. Additionally, the collected GPR data includes signatures from pipelines and manhole covers, which are also essential to identify. Figure 5 provides several examples of GPR data blocks.

Table 1: Distribution of normal and abnormal GPR data blocks by type.

| Normal | Cavity | Looseness | Crack | Pipeline | Manhole |
|--------|--------|-----------|-------|----------|---------|
| 200 | 192 | 203 | 191 | 201 | 192 |

### 4.2 Anomaly Detection of GPR Data

In subsurface anomaly detection, the initial focus involves identifying anomalies in newly collected GPR data using only previously acquired normal data free from subsurface anomaly or other objects.

---

[5]Clustering techniques, such as K-Means (Hartigan & Wong, 1979), Agglomerative clustering (Ackermann et al., 2014), or Fuzzy C-Means (Bezdek et al., 1984) can be applied based on model distance measurements.

[6]https://scikit-learn.org/

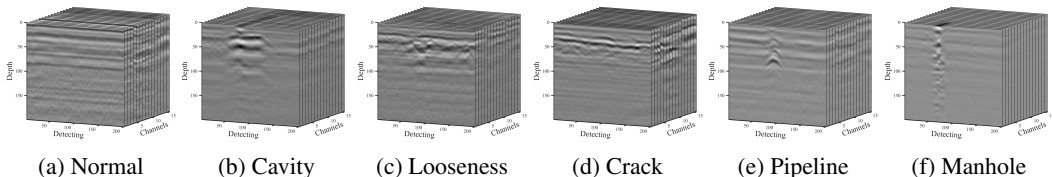

| (a) Normal | (b) Cavity | (c) Looseness | (d) Crack | (e) Pipeline | (f) Manhole |

Figure 5: Several examples of 3D GPR data blocks containing different subsurface objects.

For training, we randomly select 100 normal data blocks. For testing, we use the remaining 100 normal blocks and randomly select 20 blocks from each type of anomaly. The objective of anomaly detection is to develop a classification approach only using the training normal blocks, and then determine the normality or abnormality of each data block in the test set.

We evaluate our approach against recent baselines, which solely utilize some normal data for anomaly detection in subsequently collected data[7]: 1) Voxel-based anomaly detection methods, which treats 3D DPR data as a three-dimensional matrix: Patchcore-3D (Frolova et al., 2023), Synthetic Temporal Anomaly Guided End-to-End Video Anomaly Detection (STEAL) (Astrid et al., 2021), 3D-VAE (Brock et al., 2016), and MemAE (Gong et al., 2019). 2) Image-based anomaly detection approaches, in which 3D DPR data is considered as a multi-channel image: f-AnoGAN (Schlegl et al., 2019), and SimpleNet (Liu et al., 2023). 3) We also evaluate CuDeRes without spatial decay, denoted as CuDeRes(w/o $\mathbf{E}$), while keeping all other settings identical.

In our experiment, we exclude the experiment with the traditional LMS method that employs ESN to fit 3D GPR data blocks uni-directionally and processes the ESN readout model, primarily due to the impractical size of the ESN readout model. Fitting the data block along the detecting direction, the size of the ESN readout model is $200 \times 16 \times$reservoir size and fails to be directly classified. As for the CuDeRes, along with a comprehensive capture of multi-directional dynamics, a more compact readout model size $3 \times$reservoir size is obtained, facilitating efficient learning on the fitted models.

The results[8] are presented in Table 2. Voxel-based methods like Patchcore-3D, STEAL, 3D-VAE, and MemAE treat 3D GPR data as three-dimensional matrices, but struggle with insufficient normal data for effective optimization, failing to accurately capture normal changing information and thus diminishing their capability to distinguish variations among normal and abnormal patterns in our application. Image-based methods, such as f-AnoGAN and SimpleNet, designed for two-dimensional image format, overlook effective changing information of a certain dimension and do not adequately capture the internal dynamics in 3D GPR data. Building correlations along and among EM waves, fitting GPR data via CuDeRes adequately and accurately capturing the data-inherent multi-directional dynamics. Introducing spatial decay also improves the F1-Score by about 4%, enabling a more accurate and balanced capture of dynamics within different directions. Consequently, the CuDeRes models, fitted from abnormal blocks, markedly differ from those derived from normal data. Focusing on the data-inherent changing information, our approach demonstrates superior anomaly detection performance compared to baselines, even with the given minimal normal data support.

Table 2: Comparative performance of our approach against baselines in anomaly detection.

| Methods | Precision(Pre) | Recall(Rec) | F1-Score |
|---|---|---|---|
| Patchcore-3D | 83.2% | 86.0% | 84.6% |
| STEAL | 84.8% | 82.7% | 83.7% |
| 3D-VAE | 86.0% | 84.6% | 85.3% |
| MemAE | 79.2% | 81.6% | 80.4% |
| f-AnoGAN | 81.6% | 79.8% | 80.7% |
| SimpleNet | 80.9% | 79.3% | 79.6% |
| CuDeRes(w/o $\mathbf{E}$) | 87.9% | 87.3% | 87.6% |
| Our Approach | **92.2%** | **91.7%** | **91.9%** |

Subsequent experiments further reveal that our approach not only distinguishes between normal and abnormal models but also enables effective clustering in the CuDeRes model space, reflecting the distinct dynamics within various anomaly types.

---

[7]To our knowledge, no existing research specifically tailored for 3D GPR data performs anomaly detection on newly collected data with only limited normal data training. These baselines are able to process and analyze 3D GPR data, despite not being specially designed for it.

[8]All evaluated methods show a standard deviation of less than 2% across 5 repeats, indicating stable results.

## 4.3 THE MODEL/FEATURE CLUSTERING RESULTS

After identifying abnormal models, our approach employs clustering to categorize these models, grouping data blocks originating from the same type of anomalies. The introduced CuDeRes fits and captures the multi-directional dynamics within the GPR data without offline iterative training. The fitted models thus serve as representations of the original data blocks.

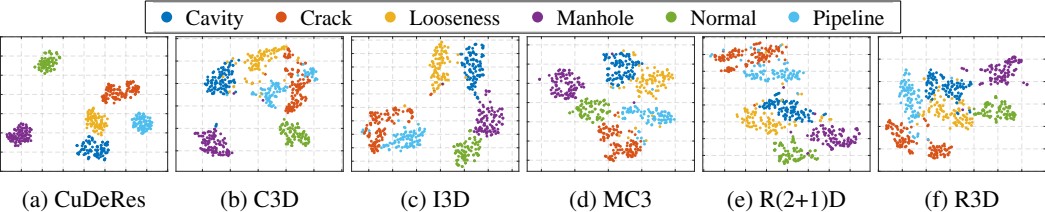

| (a) CuDeRes | (b) C3D | (c) I3D | (d) MC3 | (e) R(2+1)D | (f) R3D |

Figure 6: CuDeRes demonstrates enhanced category discrimination: 1) Marked separation between normal and abnormal models ensures reliable anomaly detection; 2) Smaller distances within classes and larger distances between classes allow for effective clustering of different anomaly types.

We use feature extractors pre-trained on the Kinetics dataset (Carreira & Zisserman, 2017), including 3D Convolutional Neural Networks (C3D) (Tran et al., 2015), 3D ResNet (R3D) (Hara et al., 2017), (2+1)D Convolutional Networks (R(2+1)D) (Tran et al., 2018), Mixed Convolutions 3D and 2D (MC3) and Inflated 3D ConvNet (I3D) (Carreira & Zisserman, 2017) on the identified abnormal blocks. These models/features are clustered by three widely used clustering algorithms: K-Means, Agglomerative clustering (AC), and Fuzzy C-Means (FCM).

Table 3 shows that CuDeRes outperforms others in anomaly clustering effectiveness. Using t-SNE (Van der Maaten & Hinton, 2008), we visualized the fitted models and other features in 2D space (Figure 6). Each point corresponds to a fitted model or feature associated with a GPR data block. Although image-like, GPR data essentially represents EM wave collections, and exhibits unique dynamics due to variations among and along EM waves caused by different underground anomalies. Pre-trained deep neural networks, typically designed for visual feature extraction, struggle to effectively

Table 3: The models/features clustering results: Accuracy (Acc), Adjusted Rand Index (ARI), and Normalized Mutual Info (NMI).

| Methods | K-Means | | | AC | | | FCM | | |
|---|---|---|---|---|---|---|---|---|---|
| | Acc | ARI | NMI | Acc | ARI | NMI | Acc | ARI | NMI |
| C3D | 0.57 | 0.36 | 0.55 | 0.69 | 0.57 | 0.67 | 0.64 | 0.46 | 0.60 |
| MC3 | 0.85 | 0.72 | 0.76 | 0.93 | 0.83 | 0.85 | 0.85 | 0.74 | 0.77 |
| R(2+1)D | 0.76 | 0.58 | 0.68 | 0.81 | 0.64 | 0.74 | 0.60 | 0.44 | 0.57 |
| I3D | 0.74 | 0.69 | 0.79 | 0.76 | 0.72 | 0.83 | 0.75 | 0.69 | 0.78 |
| R3D | 0.82 | 0.62 | 0.67 | 0.73 | 0.61 | 0.72 | 0.83 | 0.65 | 0.69 |
| CuDeRes | **0.96** | **0.91** | **0.90** | **0.97** | **0.93** | **0.93** | **0.95** | **0.89** | **0.91** |

capture and distinguish such dynamics. CuDeRes focuses on the data-inherent changing information, thoroughly capturing its multi-directional dynamics by establishing connections within the data block in multiple directions. Owing to the distinct dynamics captured, the models derived from CuDeRes exhibit superior clustering performance on different types of subsurface anomalies.

## 5 CONCLUSION

This study introduces learning in the Cubic Decay Reservoir Network (CuDeRes) for anomaly detection within 3D GPR data, specifically: 1) We introduce a novel CuDeRes, featuring three reservoirs and spatial decay in each direction, designed to adequately and accurately capture the multi-directional dynamics within GPR data, resulting in a compact fitted readout model; 2) Representing GPR data using the fitted CuDeRes model, along with a computable distance measurement between models, enables effective anomaly detection within the category-discriminative CuDeRes model space; models fitted from different anomalies show noticeable differences owing to the distinct dynamics captured, while same-type data blocks derive similar fitted models due to their consistent inherent dynamics; 3) Our approach concentrates on the inherent dynamics present in GPR data, and leverages only limited normal GPR data, easily obtainable in the detecting area, to support the subsequent anomaly detection, enabling its practical usability. Our future work will focus on reconstructing anomaly regions for intuitive visualizations of target anomalies, aiding repair efforts.

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

# A APPENDIX

## A.1 DISTANCE MEASUREMENT BETWEEN MODELS

Following the data fitting through CuDeRes, the distance between the fitted models should be defined. A possible choice is to identify parameterized models with their parameter vectors, but this makes further learning highly dependent on the specific model parameterization used. In this paper, the $p$-norm distance between models $f_1(\mathbf{x})$ and $f_2(\mathbf{x})$ is adopted:

$$\mathcal{D}_p(f_1, f_2) = \left( \int_C \|f_1(\mathbf{x}) - f_2(\mathbf{x})\|^p d\mu(\mathbf{x}) \right)^{1/p}. \tag{12}$$

where $\mu(\mathbf{x})$ is the probability density function of the input domain, and $C$ is the integral range. In this paper, we adopt $p = 2$ and firstly assume $\mathbf{x}$ is uniformly distributed. Supposing two GPR data blocks are fitted by CuDeRes, and each reservoir in the CuDeRes has $N$ neurons, the two readout models could be represented by:

$$\begin{cases} f_1(\mathbf{x}) = \mathbf{W}_1^{\text{out}}\mathbf{x} + \beta_1, \\ f_2(\mathbf{x}) = \mathbf{W}_2^{\text{out}}\mathbf{x} + \beta_2. \end{cases} \tag{13}$$

where $\mathbf{W}_1^{\text{out}}, \mathbf{W}_2^{\text{out}} \in \mathbb{R}^{1 \times 3N}$, and $\mathbf{x} \in \mathbb{R}^{3N \times 1}$.

Substituting Equation 13 into Equation 12, it could be obtained that:

$$\begin{aligned} \mathcal{D}_2(f_1, f_2) &= (\int_C \|f_1(\mathbf{x}) - f_2(\mathbf{x})\|^2 d\mathbf{x})^{1/2} \\ &= (\int_C \|(\mathbf{W}_1^{\text{out}} - \mathbf{W}_2^{\text{out}})\mathbf{x} + (\beta_1 - \beta_2)\|^2 d\mathbf{x})^{1/2} \\ &= (\int_C \|\mathbf{W}_{12}^{\text{out}}\mathbf{x}\|^2 + 2\beta_{12}\mathbf{W}_{12}^{\text{out}}\mathbf{x} + \beta_{12}^2 d\mathbf{x})^{1/2}, \end{aligned} \tag{14}$$

where $\mathbf{W}_{12}^{\text{out}} = \mathbf{W}_1^{\text{out}} - \mathbf{W}_2^{\text{out}}$, $\beta_{12} = \beta_1 - \beta_2$. Here, $f_1$ and $f_2$ are simplified representations of $f_1(\mathbf{x})$ and $f_2(\mathbf{x})$.

Note that for any fixed $\beta_{12}$ and $\mathbf{W}_{12}^{\text{out}}$, there is

$$\int_C \beta_{12}\mathbf{W}_{12}^{\text{out}}\mathbf{x} d\mathbf{x} = 0 \tag{15}$$

in the integral range $C = [-1, 1]^{3N}$.

Therefore

$$\begin{aligned} \mathcal{D}_2(f_1, f_2) &= (\int_C \|\mathbf{W}_{12}^{\text{out}}\mathbf{x}\|^2 + \beta_{12}^2 d\mathbf{x})^{1/2} \\ &= (\frac{2^{3N}}{3} \sum_{j=1}^{3N} w_j^2 + 2^{3N}\beta_{12}^2)^{1/2}, \end{aligned} \tag{16}$$

where $w_j$ is the $(1, j)$th element of $\mathbf{W}_{12}^{\text{out}}$.

We then scale of the squared model distance $\mathcal{D}_2(f_1, f_2)$ by $2^{-3N}$, and obtain

$$\mathcal{D}_2(f_1, f_2) \propto \frac{1}{3} \sum_{j=1}^{3N} w_j^2 + \|\beta_{12}\|^2 = \frac{1}{3}\|\mathbf{W}_1^{\text{out}} - \mathbf{W}_2^{\text{out}}\|^2 + (\beta_1 - \beta_2)^2, \tag{17}$$

serving as the directly measured distance between two models, enabling the utilization of distance-based learning algorithms on these models.

## A.2 INTRODUCTION OF THE GPR DATASET

In our experiments, three major subsurface diseases appear: cavities, looseness, and cracks

- "Cavity" beneath experimental roads refers to a space predominantly filled with air. These cavities can vary in shape and may have either smooth or rough boundaries. They are a primary factor leading to the subsidence observed in urban roadways. Due to the significantly lower dielectric constant of air compared to surrounding materials, EM waves encountering these cavities produce a pronounced reflection. Furthermore, these EM waves can undergo multiple reflections within the cavity, occasionally accompanied by diffraction effects.

- "Looseness" describes a soil condition characterized by increased porosity and reduced density, especially when compared to adjacent soil with comparable water content. This state leads to diminished particle cohesion, rendering the soil structure less compact. Such a scenario can potentially lead to soil collapse and subsequent formation of underground cavities. Essentially, loose soil can be conceptualized as a mixture of soil and air. The heterogeneity in soil properties within areas of looseness results in erratic reflected waves in GPR data, as waves traverse through these zones.

- "Crack" in road infrastructures refers to horizontal gaps within the previously dense underground layers, typically air-filled. These gaps create a distinct scenario where wave reflections are observed as the waves encounter the delamination. This interaction results in notable alterations and sharp discontinuities in the waveform. When detecting such horizontal delaminations, the collected waves exhibit changes nearly simultaneously, indicative of this specific underground anomaly.

In addition, our collected GPR data also encompasses signals influenced by pipelines and manhole covers. These types are also considered as "Anomaly" in our experiments. To display the GPR data more intuitively, Figure 7 shows examples of image-format single-channel GPR data. Figure 5 in our paper gives some examples of the 3D GPR data blocks collected by multi-channel GPR.

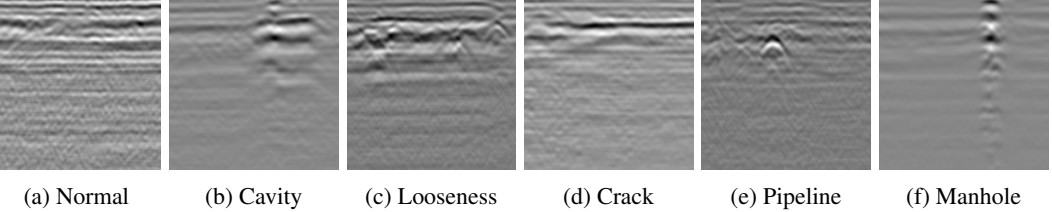

|   (a) Normal   |   (b) Cavity   |   (c) Looseness   |   (d) Crack   |   (e) Pipeline   |   (f) Manhole   |

Figure 7: Several examples of different subsurface conditions in the single-channel GPR data.

## A.3 MANIPULATIONS ON THE RAW GPR DATA

We have executed specific manipulations on the 3D GPR data to eliminate noise and accentuate subsurface objects. This process involves three key tasks: removing undesired surface echoes, reducing noise, and offsetting propagation losses. Initially, reflectance from the ground surface is eliminated. Subsequently, a standard median filter is employed on the data to diminish electromagnetic noise and interference. The final step involves applying a time-variant gain adjustment to counteract the propagation losses due to medium attenuation and the dispersion of signal energy radially. Our approach to gain manipulation involves two main stages:

- Considering that targets like manhole covers can lead to data oversaturation, we isolate these sections by setting a threshold for wave intensity significantly above the average. Post this, we apply gain adjustments to the rest of the data.

- The time-variant gain technique is utilized to progressively increase the gain exponent relative to the depth of detection, thus amplifying variations without causing oversaturation. In instances where oversaturation is observed in data collected from roads, we modify the gain for all corresponding data, ensuring the preservation of variable information. This fine-tuning process requires manual intervention.

These manipulations are designed to maintain and highlight the dynamic information in the data while avoiding oversaturation. All adjustments are conducted using the Matgpr Matlab package,

specifically tailored for GPR data analysis. It's crucial to note that these modifications are consistently applied across data from similar underground environments to ensure uniform treatment of all data gathered from the same road.

## A.4 Analysis About the Size of Data Block

In discussing the block size in GPR data analysis, it should be noted that the depth and width are primarily determined by the GPR system used. Our focus here is on the block length in the detection direction. 1) The CuDeRes treats the GPR data within a block as a unified entity, fitting the data and capturing its internal dynamics. This process is akin to ESN and other reservoir computing techniques, necessitating iterative processes in CuDeRes to fully capture the GPR data dynamics and to ensure model stability. Consequently, the block length should not be too short. 2) An overly long block may result in an excessively large range of detected abnormalities. Additionally, a larger block size can increase the iterations required for fitting. Also, the block length selection may also take into account the accuracy of the positioning device for further localizing the detected anomaly.

Anomaly detection tests on various block lengths, detailed in Table 4, demonstrate the effectiveness of our approach that remains stable across different block sizes. This stability can be attributed to CuDeRes's comprehensive capture and distinction of the data-inherent multidirectional dynamics. Adjusting the block size to suit the specific requirements of a given scenario is unlikely to compromise the effectiveness of our method.

Table 4: The Anomaly Detection Result with Different Block Length

| Block Length | Precision | Recall | F1-score |
|---|---|---|---|
| 100 | 89.92% | 90.18% | 90.05% |
| 200 | 92.23% | 91.71% | 91.92% |
| 300 | 90.85% | 89.23% | 90.03% |
| 400 | 88.68% | 89.23% | 88.95% |

## A.5 Discussion About the Reservoir Size in CuDeRes

Similar to other reservoir computing networks, like ESN, the reservoir size in CuDeRes plays a role in performance. In line with other methods involving reservoir computing, it is observed that larger reservoirs typically enhance the fitting accuracy to capture more complex dynamics. However, there are a few considerations: 1) Upon exceeding a specific threshold, the enhancement in fitting accuracy gained by increasing the reservoir size becomes incrementally minimal; 2) Larger reservoirs may result in a longer fitting process due to more iterative paths, although it's noteworthy that the fitting process in CuDeRes typically requires only a single iteration.

Table 5: The Anomaly Detection Result with Different Reservoir Size

| Reservoir Size | Precision | Recall | F1-score |
|---|---|---|---|
| 5 | 86.43% | 82.25% | 84.29% |
| 10 | 90.71% | 90.08% | 90.39% |
| 30 | 92.01% | 91.72% | 91.86% |
| 50 | 92.23% | 91.71% | 91.92% |
| 70 | 91.90% | 91.39% | 91.59% |
| 90 | 91.74% | 91.22% | 91.42% |

To evaluate the impact of reservoir size on CuDeRes's performance, we conducted tests with various sizes, the results of which are presented in Table 5. These findings indicate that reservoir sizes below 10 are not very effective. However, choosing a reservoir size around 50 tends to yield relatively stable and satisfactory results.

