# OpenReview forum: "From Data to Model: Anomaly Detection of 3D GPR Data in CuDeRes Model Space"
_ICLR.cc/2025/Conference — ICLR 2025 Conference Withdrawn Submission_

### Official Review · Reviewer_sR1J · 2024-10-26

**Soundness:** 2
**Presentation:** 2
**Contribution:** 2
**Rating:** 5
**Confidence:** 4

**Summary:**

This is paper about GPR anomaly detection. GPR signals are noisy and there lacks negative training data in some cases. So the paper proposed a method that is trained on normal data to construct a learning in the model space called LMS. Each direction of GPR measurements, i.e., x, y, z corresponds to one type of model in the LMS. After training, the method is tested on normal and abnormal GPR signals and showed better performance than other methods. Contributions include the introduction of Cubic Decay Reservoir Network (CuDeRes) and better performance in testing.

**Strengths:**

Strengths include the introduction of CuDeRes to model normal GPR data and better performance in testing and comparisons versus Patchcore-3D, STEAL, 3D VAE, and MemAE, as well as the capability to tell apart different types of buried objects such as manholes and pipes by either K-Means, Agglomerative clustering, or Fuzzy C-Means.

**Weaknesses:**

While CuDeRes model had better performance, it seems the novelty of this model is not particularly high. The example GPR signals as shown in the paper appear to have quite distinct presentation from the normal case and from each other, thus it seems the problem is not very challenging for a deep learning model to detect.
There lacks detailed description about the GPR data, adding to the difficulty to understand the paper.
In some places, the writing is a little bit difficult to follow and there are typos. For example, "U is a specific data blocks", should be "block".

**Questions:**

In modeling the three directions of GPR signals, it is understandable that there could be signal decay along the in-depth direction of the signal to the soil (the z direction according to Figure 4), but it is not clear why there should be signal decay along the other two directions. And even if so, why the decay coefficient theta is the same over all three directions in equation 3?
If seems there is a segmentation step as alluded in equation 10, how is segmentation done?
As the GPR moves down the track, is there any overlap in the data block in the method?
It is not clear why equation 15 equals to zero? Does it mean W_1^out and W_2^out is everywhere equal? If so, why? If not so, then can authors elaborate why equation 15 is zero?
It seems that each direction in 3D is modeled separately and has their own model reservoir, if that is so, what happens in testing if at location, the GPR data is a good fit, for example, for x-, y- and z-direction models but these "good" models are not from the same data point in the training process, rather the fitted x-, y- and z-direction models were from different data points in the training set. Will that cause a problem for the whole method to process?

---

> ### Author Response · Authors · 2024-11-15
> **Response to Reviewer’s Questions on Spatial Decay, Segmentation, Integral Property, and Model Fitting**
>
> Thank you very much for your review.
>
> 1) On the spatial decay mechanism:
> The introduced spatial decay is intended to account for the distances between neighboring points when constructing their relationships. Closer points are assigned stronger connections, while farther points have weaker connections. Spatial decay is not designed to simulate signal attenuation. Signal attenuation is instead addressed through gain adjustments, as described in the Appendix.
>
> 2) On the segmentation process:
> The data is segmented into fixed-size blocks that slide across the dataset, with overlapping allowed between adjacent blocks. This ensures that no anomalies are missed and provides comprehensive coverage of the anomalous regions.
>
> 3) On the zero integral in Equation 15:
> Equation 15 is zero because it represents the integral of an odd function of \(x\) over the symmetric range \([-1, 1]\). This property holds regardless of whether \(W_1^{\text{out}}\) and \(W_2^{\text{out}}\) are equal. There is no connection between the two conditions.
>
> 4) On the fitting and use of reservoirs in testing:
> It seems there is a misunderstanding of our approach. When fitting the data, the relationships constructed consider each point and its neighboring points in the three directions. The fitted model represents the entire data block and maps it into the model space. The question regarding "good" models from different points does not apply to our method because the model representation is for the entire data block, not individual points.

---

### Official Review · Reviewer_Gw4v · 2024-10-27

**Soundness:** 3
**Presentation:** 3
**Contribution:** 3
**Rating:** 6
**Confidence:** 2

**Summary:**

This paper presents a novel method, called cubic decay reservoir network, for anomaly detection within 3D Ground Penetrating Radar data.

**Strengths:**

The method seems interesting and the paper is well written with comprehensive experiments that demonstrate the relevance of the proposed approach.

**Weaknesses:**

The proposed method is motivated by the objective of diminishing the computational demands, as opposed the 3D-CNNs necessitating considerable computational resources. However, not much is said on the computational complexity of the proposed method, and the authors need to provide results on the comparative analysis.

**Questions:**

What about the computational complexity ?

---

> ### Author Response · Authors · 2024-11-13
> **Thank you very much for recognizing our work.**
>
> Thank you very much for recognizing our work.
>
> The fitting of CuDeRes for each data block only requires a single ridge regression (see Equation 5) and does not involve iterative training, which is a key advantage of reservoir computing. Moreover, the distance metric between readout models is directly solvable, enabling our method to perform well in real-time and to scale effectively on large datasets.

---

> > ### Comment · Reviewer_Gw4v · 2024-11-17
> >
> > We thank the authors for their reply.
> >
> > However, it is still hypothetical that the proposed method performs well in real-time. It would have been more relevant to quantify the computational cost and to demonstrate empirically its relevance in a real-time implementation, while comparing it to related work.

---

### Official Review · Reviewer_HsFY · 2024-11-01

**Soundness:** 2
**Presentation:** 3
**Contribution:** 2
**Rating:** 3
**Confidence:** 4

**Summary:**

The paper addresses the problem of anomaly detection in images,
esp. for multichannel ground penetrating radar images (called
3d-GPR). The authors propose to characterize such an image by
the parameters of a ridge regression that aims to reconstruction
a blurred version from a pixel and its three left, upper and behind
neighbors. In an experiment on a GPR dataset with several classes,
they show that their method outperforms other methods
for image anomaly detection.

**Strengths:**

- s1. interesting data modality: 3d GPR data.
- s2. interesting progressive blurring operation: project each pixel
  to a combination of already blurred neighbors.
- s3. well written.

**Weaknesses:**

- w1. irreproducible: experiments on a single, non-public dataset.
- w2. major baselines missing in the experiments, e.g. Liu et al. 2024
  and Zhou et al. 2024.
- w3. simple ablations missing, e.g., blurring pixels based on their
  neighbors' observed intensities.

The paper addresses an interesting, not so heavily researched
data modality: ground penetrating radar images. It makes some
interesting, specific proposals. But currently I see three major
issues:

w1. irreproducible: experiments on a single, non-public dataset.
- Without showing a performance against other methods on a
  public dataset, it will be very difficult for other researchers
  to compare against the proposed method. It also is more
  or less impossible to assess the quality of the method.
- The authors also do not release any code to make such a
  reproduction easier.

w2. major baselines missing in the experiments, e.g. Liu et al. 2024
  and Zhou et al. 2024.
- The authors mention these two papers in their related work
  as being close and GPR-specifc. Can you compare agains them?

w3. simple ablations missing, e.g., blurring pixels based on their
  neighbors' observed intensities.
- Your method is specific in the sense that your blurring operation
  in eq. 1 blurs progressively: the value of the next pixel depends
  on the already blurred values of previous pixels. This leads to
  a very deep architecture. Can you compare on a simple setup
  where you blur on just the observed values of the neighbors?
- You fix theta to 1 and test only against theta = 0 ("w/o E").
  Can you run a full parameter sensitivity analysis for theta?

There also are some smaller issues that the authors could
look into:
- i1. Your model does not depend on x_a, y_b and z_c, but just
  on x_{a+1}-x_a (in eq. 3). You could simplify your representation
  by just using indices a, b, c and say what scaling factors you
  use in eq. 3.
  - Can you specify what these 3 numbers,  x_2-x_1, y_2-y_1 and z_2-z_1
    are in your experiment?
  - line 417 you say the "width" of a block is 2.5m. Earlier you said
    your third channel is the antenna channel (e.g., fig. 2). Can you clarify?
- i2. line 293 you characterize the task as "next point prediction".
  But in your case u(x_a, y_b, z_c) is one of the inputs. So would
  it not be more appropriate to call it a reconstruction (from a
  blurred input)?
- i3. line 330 "\mu(x) denotes the probability density over the input
  domain" one easily could read as if you mean the ground truth
  distribution of the GPR images. But then you clarify to assume
  a uniform distribution.
- i4. The conditions on the random parameters W^x, W^y, W^z
  are formulated only vaguely as "satisfy the Echo State Property"
  (line 269). This could be made self-contained.

**Questions:**

- q1. Can you demonstrate your method on any public datasets,
  preferably some used already for anomaly detection for images?
- q2. How does your method compare against the methods from
  Liu et al. 2024 and Zhou et al. 2024 mentioned in your related work?
- q3. If in your blurring operation (eq. 1), you blur not based on the already
  blurred values of the neighbors, h*, but on the observed values
  (u(x_{a-1}, y_b, z_c), u(x_a, y_{b-1}, z_c), u(x_a, y_b, z_{c-1})),
  does this (way less deep, simpler) model work as well?

---

> ### Author Response · Authors · 2024-11-13
> **Response to Reviewer’s Feedback on Reproducibility, Baselines, and Methodological Clarifications**
>
> Thank you for thoroughly reviewing our work and providing valuable feedback.
>
> w1. Reproducibility of the Method: We have provided demo code and examples in the supplementary material. The proposed method is fully reproducible.
>
> w2. Major Baselines Missing in the Experiments: The methods by Liu et al. and Zhou et al. (D3 model and CubeRes) rely on training and optimization based on data from different anomaly categories. In contrast, our work uses only normal data for anomaly detection, making it a different task and therefore not directly comparable.
>
> w3. Simple Ablations Missing: Our CuDeRes approach is constructed using reservoir computing, where connections between neighboring points are built through hidden states rather than direct associations in the original data. We conducted ablation studies and discussions, including removing the spatial decay mechanism (shown as “CuDeRes(w/o E)” in Table 2), and discussed single-direction ESN fitting on lines 453-458 of page 9. Due to the large dimensionality of the readout model, single-direction ESN fitting cannot be applied to 3D GPR data.
>
> i1. Specification of Values for x₂ - x₁, y₂ - y₁, and z₂ - z₁: These represent the distances between neighboring points along three spatial directions. Regarding the width of a block (2.5m), this corresponds to the distance from the first to the 16th channel, meaning X2−X1=2.5/16X_2 - X_1 = 2.5/16X2−X1=2.5/16.
>
> i2. Rename “Next Point Prediction” to “Reconstruction”: This is a good suggestion, and we agree with the change.
>
> i3. Clarification on the Probability Density (Line 330): There is no direct relationship between the probability density and the ground truth distribution of GPR images. We apologize for any confusion caused by this wording.
>
> i4. Conditions on Random Parameters W^x, W^y, and W^z: The Echo State Property (ESP) is a condition in reservoir computing to maintain system stability. It is not unique to our work or a primary contribution of this paper. We apologize for not making this clearer.

---

### Official Review · Reviewer_DtN8 · 2024-11-03

**Soundness:** 3
**Presentation:** 3
**Contribution:** 2
**Rating:** 5
**Confidence:** 3

**Summary:**

This paper presents a framework for anomaly detection in 3D Ground Penetrating Radar (GPR) data using the Cubic Decay Reservoir Network (CuDeRes). CuDeRes fits 3D GPR data segments into model representations, which are stored in a "Model Depot" as references for normal data patterns. The approach detects anomalies by comparing new data representations with stored models, leveraging multi-directional spatial decay to capture subsurface dynamics. Experiments demonstrate the model’s effectiveness in identifying subsurface anomalies, achieving an F1-score of 91.9% and surpassing baseline methods.

**Strengths:**

1. **Efficient Use of Limited Normal Data**: The framework relies on a small amount of normal data, making it feasible for real-world applications where labeled anomaly data is scarce.

2. **Novel Application of Multi-Directional Reservoirs**: By incorporating three reservoirs with spatial decay, the CuDeRes architecture effectively captures multi-directional dynamics, essential for processing complex 3D GPR data.

3. **Practical Relevance for Subsurface Anomaly Detection**: This model has clear potential applications in structural health monitoring, especially for roads and urban infrastructure.

4. **Comprehensive Experimental Validation**: The model is rigorously tested across various anomaly types and compared to multiple baseline models, demonstrating both its classification and clustering performance.

**Weaknesses:**

1. **Robustness in Anomaly Typing**: The clustering step for anomaly type identification is not extensively evaluated. The paper would benefit from additional metrics or case studies demonstrating the accuracy and practical impact of identifying specific anomaly types.

2. **Scalability**: The approach’s scalability to larger datasets or real-time applications is not well-explored. Given the model's complexity, its feasibility in high-throughput scenarios is uncertain.

3. **Distance Metric**: The paper uses the p-norm distance for model comparisons without exploring alternative metrics or adaptive methods. This approach may limit its effectiveness in distinguishing anomalies with subtle variations from normal models.

4. **Manual Parameter Tuning**: The CuDeRes model performance depends on manually set parameters (e.g., reservoir size, spatial decay rate) without clear guidelines on optimal selection. This could limit the method's applicability to other data sources or detection tasks without significant retraining or adjustment.

**Questions:**

1. **Scalability and Computational Efficiency**: Given the complex nature of fitting CuDeRes on each GPR data block, what is the computational cost for large-scale data processing? Has the model been tested in a real-time setting, and if so, how does it perform?

2. **Robustness**: How accurate is the clustering in differentiating between types of anomalies? Are there specific situations where the model fails to distinguish between similar anomaly types, such as looseness versus cavities?

3. **Distance Measures**: Why was p-norm distance selected as the primary comparison metric? Would a learned distance metric improve the model's sensitivity to different anomaly patterns without impacting generalizability?

4. **Parameter Sensitivity**: The paper lacks an in-depth analysis of how CuDeRes's performance is affected by parameter variations. Have the authors considered any methods for dynamic parameter tuning, or is the current approach limited to the manually selected parameters?

5. **Ablation Study**:  Given the complexity of the CuDeRes model, why did the authors not conduct an ablation study to examine the contributions of individual components? For instance, how does the model perform without the spatial decay mechanism or when using single-direction fitting instead of multi-directional fitting? Additionally, would testing alternative distance metrics or varying reservoir sizes provide insights into which design choices are most critical for effective anomaly detection?

---

> ### Author Response · Authors · 2024-11-13
> **Response to Reviewer’s Comments on Scalability, Robustness, Distance Measures, and Ablation Studies**
>
> We appreciate the reviewer’s recognition of our work. Below are our responses to the raised points:
>
> Scalability and Computational Efficiency: CuDeRes fitting for each data block requires only a single ridge regression (see Equation 5), with no iterative training needed. This is a key advantage of reservoir computing, allowing for efficient fitting. Additionally, the model distance metrics are directly solvable, ensuring that our method is feasible for real-time applications and scalable to large datasets.
>
> Robustness in Anomaly Typing: The clustering results for CuDeRes are presented in Figure 6 and Table 3. To further verify the model’s discriminative ability across various anomaly categories, we applied three different clustering methods, achieving robust performance in anomaly type clustering.
>
> Distance Measures: As the CuDeRes readout model is not a vector, Euclidean distance is unsuitable for direct application. Previous studies in model space-based approaches have validated the effectiveness of the p-norm distance. Given the limited normal data available in our anomaly detection context, optimizing the distance metric would be challenging due to insufficient data support.
>
> Parameter Sensitivity: We have provided details on the primary parameter settings and ranges in the Experiment section and the Appendix, along with necessary discussions.
>
> Ablation Study: We conducted ablation studies. The results without the spatial decay mechanism (“without the spatial decay mechanism”) are shown as “CuDeRes(w/o E)” in Table 2. Results for “single-direction fitting” are discussed in lines 453-458 on page 9, where, due to the high dimensionality of the readout model, single-direction ESN fitting is impractical for 3D GPR data.

---

### Note · Authors · 2024-11-17

I have read and agree with the venue's withdrawal policy on behalf of myself and my co-authors.